# Tumor Response and Its Impact on Treatment Failure in Rectal Cancer: Does Intensity of Neoadjuvant Treatment Matter?

**DOI:** 10.3390/cancers16213673

**Published:** 2024-10-30

**Authors:** Markus Diefenhardt, Daniel Martin, Maximilian Fleischmann, Ralf-Dieter Hofheinz, Michael Ghadimi, Claus Rödel, Emmanouil Fokas

**Affiliations:** 1Department of Radiotherapy and Oncology, University Hospital, Goethe University Frankfurt, 60590 Frankfurt, Germany; martin@med.uni-frankfurt.de (D.M.); fleischmann@med.uni-frankfurt.de (M.F.); croedel@med.uni-frankfurt.de (C.R.); emmanouil.fokas@uk-koeln.de (E.F.); 2Frankfurt Cancer Institute (FCI), University Hospital, Goethe University Frankfurt, 60590 Frankfurt, Germany; 3Partner Site Frankfurt, German Cancer Research Center (DKFZ) and German Cancer Consortium (DKTK), 60590 Frankfurt, Germany; 4Department of Medical Oncology, University Hospital Mannheim, University Heidelberg, 68167 Mannheim, Germany; ralf-dieter.hofheinz@medma.uni-heidelberg.de; 5Department of General, Visceral and Pediatric Surgery, University Medical Center, University Göttingen, 37099 Göttingen, Germany; mghadimi@med.uni-goettingen.de; 6Department of Radiation Oncology, Cyberknife and Radiation Therapy, Faculty of Medicine and University Hospital Cologne, University of Cologne, 50937 Köln, Germany

**Keywords:** rectal cancer, adjuvant chemotherapy, downstaging, total neoadjuvant treatment

## Abstract

The response to neoadjuvant treatment in rectal cancer varies among patients, while the evidence of benefit from additional adjuvant chemotherapy, especially in patients with persistent tumor and/or lymph node metastases, is unclear. In our cohort study of 1778 patients, we showed that persistent tumor and/or lymph node metastasis after intensified neoadjuvant treatment, e.g., total neoadjuvant treatment (TNT), was not more strongly associated with an extensive risk of treatment failure (TF, local recurrence and/or distant metastasis) than after less intensive neoadjuvant treatment, e.g., 5-flurouracil-based chemoradiotherapy (5-FU CRT). These results may help clinicians to individualize their surveillance strategies, but do not provide an argument for adding additional chemotherapy in non-responders after TNT.

## 1. Introduction

The multimodal treatment of locally advanced rectal cancer has evolved over the past decades. The introduction of neoadjuvant radiotherapy, neoadjuvant chemoradiotherapy (CRT), and total mesorectal excision (TME) has improved locoregional control [1,2,3]. The intensification of neoadjuvant CRT and/or adjuvant CT with the addition of oxaliplatin provided inconsistent results [4]. More recently, total neoadjuvant treatment (TNT) led to enhanced pathologic complete response rates, improved disease-free survival (DFS), and, partly, improved overall survival (OS) [5,6,7]. Nevertheless, ypN+ was reported in 25% of patients treated with TNT compared to 32% in the standard of care group, and in 17% of patients treated with TNT compared to 32% in the standard of care group in the RAPIDO and PRODIGE23 trials [5,6].

The management of patients with minor or no downstaging or no substantial tumor response after (intensified) neoadjuvant treatment remains challenging. Considering the inconsistent results of adjuvant chemotherapy after standard 5-FU CRT, it is unclear whether additional adjuvant chemotherapy might improve survival in these patients [8]. It has been hypothesized that patients with ypN+ after neoadjuvant treatment reflect a subgroup with aggressive, therapy-resistant tumor biology at high risk of developing local recurrence or distant metastasis [9].

Here, we examined the extent of the response to neoadjuvant treatment and its impact on long-term treatment failure, stratified by the intensity of neoadjuvant treatment. We hypothesized that, in rectal cancer patients without downstaging or with only minor tumor response despite more intensive TNT, which may reveal an extremely aggressive phenotype of rectal cancer, the incidence of treatment failure may be even higher compared to (standard) neoadjuvant CRT.

## 2. Materials and Methods

The CAO/ARO/AIO-94 trial recruited patients between February 1995 and September 2002, the CAO/ARO/AIO-04 trial between July 2006 and February 2010, and the CAO/ARO/AIO-12 trial between June 2015 to January 2018. The design and oncologic outcomes have been previously published [1,10,11,12,13,14]. Eligibility criteria for this post hoc analysis excluded patients with no curative post-surgical status (R2, ypM+, no data on surgery). With respect to the CAO/ARO/AIO-94 trial, only patients treated within the neoadjuvant CRT arm were included in this analysis. A graphical abstract of the different treatment schemes is provided as Appendix A. All sites obtained medical ethics committee approval and written patient informed consent.

Tumor response was defined by the extent of downstaging: post-neoadjuvant tumor stage according to UICC (ypTNM or ycTNM) minus pre-treatment clinical tumor stage according to UICC (cTNM): Progress: >0, No response: =0, Limited downstaging: =−1, Intermediate downstaging: =−2, Extensive downstaging: =−3. TRG was recorded in the study cohort according to Dworak et al. [15] The five-tier classification included TRG 4 (no vital tumor cells detectable), TRG 3 (only scattered tumor cells amidst fibrosis with or without acellular mucin), TRG 2 (predominant fibrosis with scattered but easily recognizable tumor cells), TRG 1 (predominant tumor with prominent fibrosis or vasculopathy), and TRG 0 (no regression) and it was grouped into a three-tier TRG system [15,16].

Treatment failure (TF) events were defined as occurrences of local recurrence or distant metastasis, whichever occurred first. The time to TF was defined as the time from curative surgery to the TF event. To note, we decided not to use disease-free survival because the definition differed between the trials, and we did not use overall survival because we have previously shown a significant difference in survival between the cohorts after treatment failure [17]. The latter highlights the limitation of OS as a clinical endpoint for the assessment of the efficacy of primary treatments in trial cohorts recruiting patients over a longer period of time. The Pearson’s chi-squared test was used to assess differences in clinical characteristics. Kruskal–Wallis tests and Dunn–Bonferroni post hoc tests were used to examine the association of neoadjuvant treatment approach with time to surgery and extent of treatment response. The “rma” function in the “metafor” package of the R statistical software was used to examine differences in the association of treatment response and risk for TF stratified by the intensity of the neoadjuvant treatment approach. This moderation effect analysis examined whether the relationship between pathological tumor stage or TRG and risk of treatment failure changed depending on the intensity of neoadjuvant treatment approach, which can help to understand if and how different neoadjuvant treatment approaches influence the effect size [18]. The median follow-up was calculated with the reverse Kaplan–Meier approach. DFS was examined with the log-rank test and incidences of TF were analyzed by Fine–Gray regression models with death as the competing risk. The proportional hazard assumptions (PHA) were tested with the “cox.zph” function in the “survival” package and revealed no PHA violations. A competing risk regression model with an interaction term was used to examine whether the effect of treatment on hazard rates depended on the level of tumor response as assessed by pathological stage or TRG, thus allowing an assessment of whether the relationship between tumor response and outcome changes with different treatments [19,20]. Analyses were performed using the R4.2.1 software. A *p*-value < 0.05 was considered significant. Data in the analyses reported here were analyzed between October 2023 and June 2024.

## 3. Results

Of the 1948 patients treated within the three trials, 19 patients were excluded because of incomplete patient data, while 151 were excluded because they did not meet the eligibility criteria of this post hoc analysis. Of the remaining 1778 patients (1254 males and 524 females; median age: 62.6 years, age range: 19–84 years) with locally advanced rectal cancer (cT3/4 or cN+), 934 were treated with 5U-CRT, 560 with concurrent 5-FU/Ox CRT +/− adjuvant CT, and 284 with TNT. Appendix A shows the flow diagram of the present analysis and the distribution of pre-treatment clinical characteristics. Surgical characteristics and pathological characteristics are reported in Table 1.

Patients treated with TNT had slightly more advanced tumor stages, whereas the distribution of age, sex, and tumor location did not differ. The median neoadjuvant treatment time was longer in patients treated with TNT (median: 139 days TNT vs. 91 days 5-FU CRT/5-FU/Ox CRT, *p* < 0.001). Surgical methods did not differ significantly between treatment approaches (*p* = 0.276), although R1 resections were slightly higher in the TNT cohort (4.1% vs. 2.2%) (Table 1).

The tumor response was significantly enhanced by 5-FU/Ox CRT compared to 5-FU CRT (*p* = 0.046) and further after TNT (*p* < 0.001), with less patients having no response or local progression (24.65% after TNT vs. 35.2% after 5-FU/Ox vs. 37.9% after 5-FU, Figure 1). Pathological parameters are summarized in Table 1.

Of all patients, 25.7% had a pCR (including 8 patients with cCR who refused surgery) after TNT vs. 18.6% after 5-FU/Ox (*p* = 0.03) vs. 12.5% after 5-FU CRT (*p* = 0.007, Figure 2).

Regardless of the neoadjuvant treatment approach, distant metastases were the predominant reason for TF (5-FU CRT: 84%, 5-FU/OX CRT: 88%, TNT: 83%). The median time to treatment failure ranged from 16 months (5-FU CRT) to 19 months (5-FU/Ox CRT) with no clinically meaningful differences in the median time to distant metastasis, locoregional recurrence, and treatment approach (Appendix A).

After a median follow-up of 55 months for the entire cohort (IQR: 37 months–62 months), the cumulative incidence of TF was significantly lower after 5-FU/Ox CRT (HR 0.78, CI 95% 0.61–0.98, *p* = 0.037) but not after TNT (HR 0.87, CI 95% 0.66–1.16, *p* = 0.40) compared to 5-FU CRT (Figure 3), while the competitive risk of death slightly increased throughout the follow-up period (Appendix A). The study cohort was further analyzed to explore the association of clinical, surgical, and pathological parameters with the risk of TF. R1 resection was associated with a higher risk only after 5-FU/Ox CRT (HR 3.24 (CI 9%% 1.28–8.22), *p* = 0.01)), whereas ypN+ and TRG 4 were associated with a higher and lower risk of TF, respectively, regardless of the neoadjuvant treatment approach. No clinical, surgical, or pathological parameter showed an inverse association with the risk of TK between the different neoadjuvant treatment approaches (Appendix A).

The extent of tumor response was significantly associated with the incidence of TF in the entire study cohort (HR no response vs. downstaged: 3.44 [CI 95% 2.75–4.30, *p* < 0.001], HR progress vs. downstaged: HR 5.00 [CI 95% 3.64–6.75, *p* < 0.001] Appendix A), and when grouped by neoadjuvant treatment approach (5-FU CRT: Appendix A, 5-FU/Ox: Appendix A, and TNT: Appendix A). However, the incidence of TF stratified by the pathological outcome (ypTNM stage) did not differ significantly between neoadjuvant treatment approaches (Figure 4). In addition, tumor response as assessed by TRG according to Dworak et al. [15] was significantly associated with the incidence of TF in the overall cohort and when grouped by neoadjuvant treatment approach (Appendix A, Appendix A), and did not differ significantly between neoadjuvant treatment approaches, either (Appendix A, Appendix A).

Moderation analyses did not reveal significant differences, after stratification by neoadjuvant treatment approach, in the incidence of TF in patients with ypT+ ypN0 (5-FU CRT: HR 2.05 (CI 95% 1.21–3.47), 5-FU/Ox CRT HR 2.24 (CI 95% 1.10–4.54) and TNT 2.20 (CI 95% 0.97–4.97), test for moderation effect: *p* = 0.98) and with ypTany ypN+ (5-FU CRT 5.08 (CI 95% 2.50–3.78), 5-FU/Ox CRT 5.59 (CI 95% 2.76–11.3), and TNT 4.86 (CI 95% 2.15–11.0), test for moderation effect: *p*= 0.96). In addition, moderation analyses did not reveal significant differences, after stratification by TRG, in the incidence of TF in patients with TRG 0/1 (5-FU CRT HR 6.40 (CI 95% 3.11–13.2), 5-FU/Ox HR 4.12 (CI 95% 1.71–9.94) and TNT HR 2.63 (1.03–6.70), test for moderation effect: *p*= 0.31) and with TRG 2/3 (5-FU CRT HR 3.69 (CI 95% 1.87–7.44), 5-FU/Ox HR 3.16 (CI 95% 1.44–6.93), and TNT HR 2.47 (1.03–6.70), test for moderation effect: *p* = 0.74).

Furthermore, in a competing risk regression model, the association between tumor response, assessed by calculated downstaging or by TRG, and risk for treatment failure, adjusted for treatment and including an interaction term between downstaging or TRG and treatment approach, confirms the significant association between downstaging or TRG and risk of TF, but neither treatment approach nor the interaction terms showed a significant association with treatment failure (Table 2 and Table 3).

## 4. Discussion

The optimal adjuvant treatment and surveillance strategy after limited tumor response in patient with locally advanced rectal cancer after various forms of neoadjuvant treatment remains unclear. Here, we examined the impact of standard 5-FU-CRT versus intensified 5-FU/Ox-CRT versus TNT on pathological outcomes and downstaging and assessed its implications for long-term TF. We did not observe that enhanced tumor response after TNT was associated with a lower incidence of TF compared to 5-FU CRT, whereas, after 5-FU/Ox CRT, enhanced tumor response correlated with a lower incidence of TF in our cohort. Time to surgery (139 days after randomization in patients treated with TNT) was significantly longer compared to 91 days after 5-FU or 5-Fu/Ox CRT. This prolonged time to surgery may also allow for increased tumor response and should be considered when interpreting the impact of intensified neoadjuvant treatment on pathological outcomes [21].

Regardless of neoadjuvant treatment intensity, tumor responses assessed by the extent of downstaging and post-neoadjuvant tumor stage were significantly associated with TF. We hypothesized that intensified neoadjuvant treatment approaches could (1) shift more patients into more favorable, early pathological stages, whereas (2) patients without downstaging may have a particularly dismal prognosis, an effect we previously described as a reverse stage migration (Will Rogers) phenomenon [9]. However, in our analysis, the relative risk for TF stratified by post-neoadjuvant tumor stage or the extent of tumor response was not significantly altered by different neoadjuvant treatment approaches. Especially in patients with ypN+ after TNT, we did not find an extensive increase in the incidence of TF compared to patients with ypN+ treated with 5-FU CRT or 5-FU/Ox CRT. Thus, we could not confirm our hypothesis.

The management of patients with positive pathological lymph nodes after neoadjuvant 5-FU +/− oxaliplatin CRT, or TNT, respectively, remains challenging. Previous analyses by our group indicated that adherence to adjuvant CT does not impact on disease-free survival when performed unstratified [4]. In a subgroup analysis of patients treated in the standard of care group (i.e., Capecitabine-based CRT) of the RAPIDO trial, which was not stratified by pathologic outcomes, adjuvant CT appeared to improve long-term oncologic outcomes as a function of treatment adherence, but this did not reach statistical significance [22]. Further evidence that patients with poor response to neoadjuvant treatment may benefit from adjuvant chemotherapy is lacking [23]. In addition, there is limited evidence on whether non-adherence to TNT might influence the benefit of adjuvant treatment or which adjuvant treatment protocol (doublets vs. triplets vs. targeted therapy) might improve patient outcomes. The ESMO guidelines (2017) recommend adjuvant CT after CRT in patients with yp stage III (and “high-risk” yp stage II), whereas the NCCN Guideline recommends adjuvant chemotherapy for patients with stage II/III rectal cancer treated with neoadjuvant chemoradiotherapy, but not following the TNT approach [24]. In the RAPIDO trial, no additional adjuvant chemotherapy was considered after TNT, irrespective of treatment response [25]. In the Prodige23 trial, even patients treated with induction mFolFIRINOX received adjuvant CAPOX/FOLFOX for 8 weeks [6].

Our study has several limitations. First, post hoc analyses and comparisons between three independent trial cohorts can be biased by a wide variety of factors. Differences in recruiting periods, treatment centers, and physicians may have led to bias. Second, inclusion and exclusion criteria were not the same in the three trials. For example, MRI-defined inclusion criteria were not required in the CAO/ARO/AIO-94 trial, which may have impacted our results even if we excluded non-curative post-surgical patients in all trials and stratified all patients based on pathological outcome. Third, a post-treatment surveillance program in all three trials was performed in accordance with the Guidelines of the German Cancer Society for colorectal carcinomas, but surveillance recommendations have changed over the years [26]. In addition, MRI or CT examinations may have been increasingly performed in recent years, particularly in high-risk patients, beyond the recommendations of the guidelines by individual physicians’ choice. Fourth, comparing the pre-treatment clinical stage with the post-treatment pathological outcome is biased by the potential of overstaging of T stage, or the inaccuracy of MRI for nodal staging [27,28]. These challenges in assessing the clinical node status in rectal cancer patients emphasize the need for better strategies in radiology staging in rectal cancer [29,30]. Fifth, TRG was not available in all patients, no central pathology review was performed, and no analysis of inter- and intra-observer variability of TRG scoring was conducted [16].

## 5. Conclusions

In conclusion, this post hoc cross-trial comparison of patients treated within three randomized clinical trials with neoadjuvant 5-FU CRT or 5-FU/Ox CRT, or TNT showed significant differences in the incidence of TF with death as a competing risk when analyzed by post-neoadjuvant tumor stage or tumor response, but no relative differences when stratified by the intensity of neoadjuvant treatment. An extensive risk of TF, and therefore the need for additional adjuvant treatment after TNT in patients with limited tumor response or ypN+ stage after TNT, cannot be inferred from the data used in this analysis. Without prospective evidence of benefit from additional adjuvant chemotherapy in these patients, we would suggest that surveillance protocols should be more individualized based on the highly significant association between pathological outcome and the incidence of TF.

## Figures and Tables

**Figure 1 cancers-16-03673-f001:**
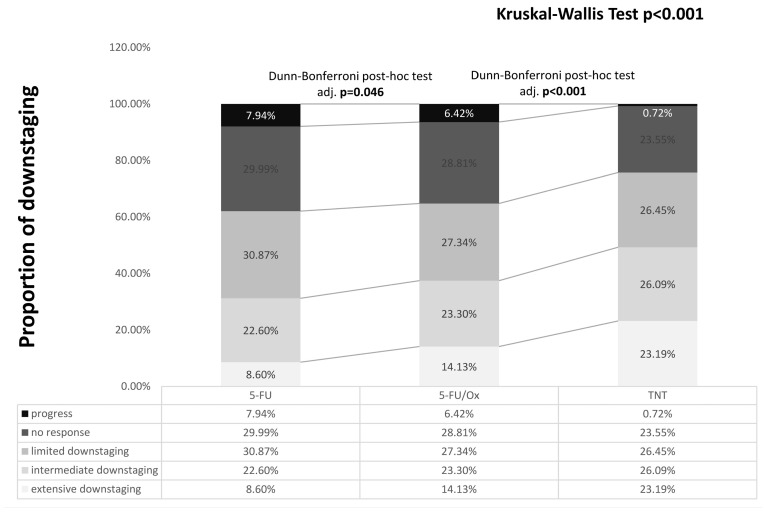
Downstaging after different neoadjuvant treatment. Differences were assessed with the Kruskal–Wallis test and Dunn–Bonferroni post hoc tests. Calculation: post-neoadjuvant tumor stage according to UICC (ypTNM or ycTNM) minus clinical baseline tumor stage according to UICC (cTNM): Progress: >0, No response: =0, Limited downstaging: =−1, Intermediate downstaging: =−2, Extensive downstaging: =−3.

**Figure 2 cancers-16-03673-f002:**
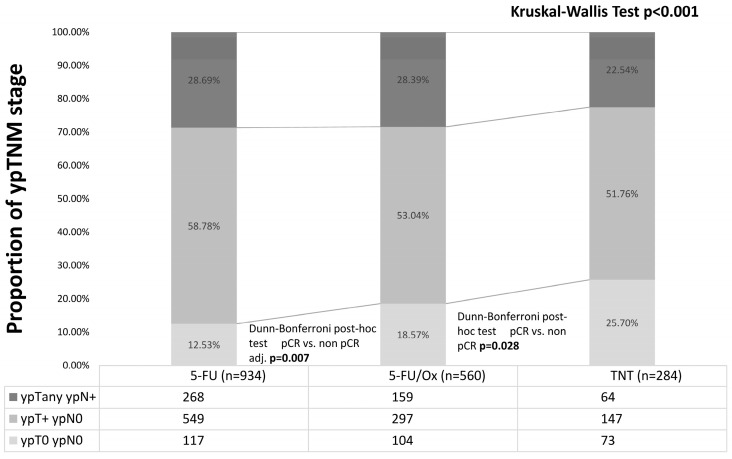
Post-neoadjuvant tumor stage after neoadjuvant 5-FU CRT, 5-FU/Ox CRT, or TNT. Differences were assessed with the Kruskal–Wall test and Dunn–Bonferroni post hoc tests.

**Figure 3 cancers-16-03673-f003:**
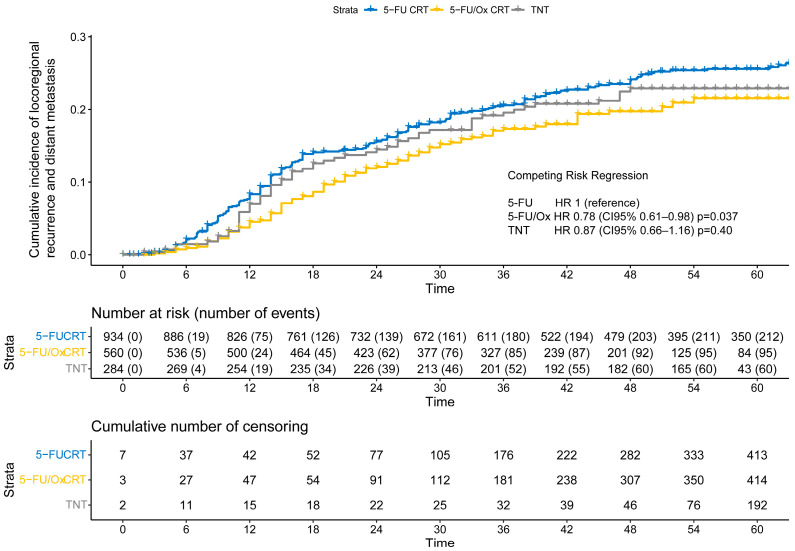
The cumulative incidence of locoregional recurrence and distant metastasis (treatment failure) after neoadjuvant 5-FU CRT, 5-FU/Ox CRT, or TNT. A competing risk regression, a model with death as the competing risk was used to assess statistical differences between treatment approaches.

**Figure 4 cancers-16-03673-f004:**
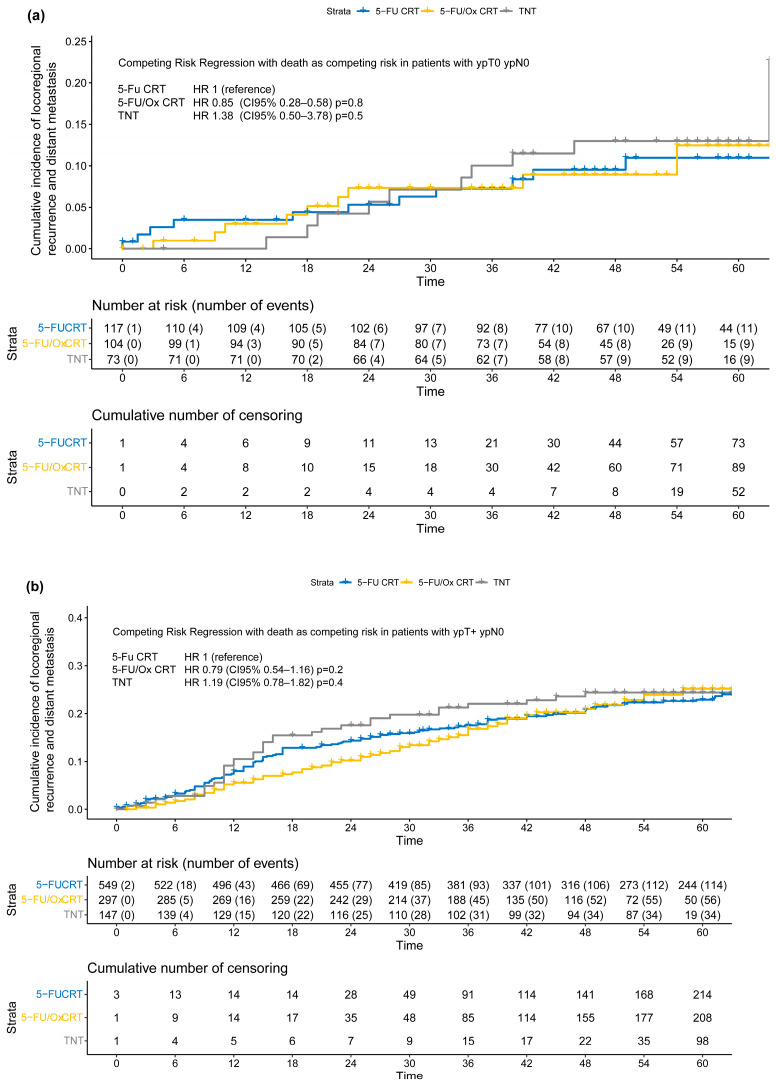
(**a**–**c**) Cumulative incidence of locoregional recurrence and distant metastasis (treatment failure) according to post-neoadjuvant tumor stage (**a**) ypT0 ypN0 or ycT0 ycN0 or (**b**) ypT+ ypN0 or (**c**) ypTany ypN+ after neoadjuvant 5-FU CRT, 5-FU/Ox CRT, or TNT. A competing risk regression model with death as the competing risk was used to assess statistical differences between treatment approaches.

**Table 1 cancers-16-03673-t001:** Distribution of clinical, surgical and pathological characteristics.

Characteristics	Study Cohort	5-FU CRT *n* = 934	5-FU/Ox CRT *n* = 560	TNT *n* = 284	*p*-Value
Gender					
Male	1254 (70.5%)	660 (70.7%)	401 (71.6%)	193 (68.0%)	
Female	524 (29.5%)	274 (29.3%)	159 (28.4%)	91 (32.0%)	0.542 *
Age					
Median age	1778 (100%)	62 years	64 years	61 years	0.084 **
Tumor localisation					
Low <0 cm to 5 cm of the anal verge	692 (39.6%)	351 (38.4%)	224 (40.5%)	117 (41.9%)	
Intermediate 5 cm to 10 cm of the anal verge	885 (50.7%)	469 (51.3%)	278 (50.3%)	138 (49.5%)	
High >10 cm of the anal verge	170 (9.7%)	95 (10.4%)	51 (9.2%)	24 (8.6%)	0.750 *
Missing	31	19	7	5	
cT					
cT2	115 (6.6%)	42 (4.9%)	19 (3.4%)	10 (3.5%)	
cT3	1501 (85.9%)	760 (88.5%)	507 (90.5%)	234 (82.4%)	
cT4	130 (7.4%)	57 (6.6%)	33 (5.9%)	40 (14.1%)	<0.001 *
Missing	75	75	1	0	
cN					
cN0	464 (26.9%)	302 (33.3%)	137 (25.1%)	25 (9.1%)	
cN+	1264 (73.1%)	605 (66.7)	408 (74.9%)	251 (90.9%)	<0.001 *
Missing	50	27	15	8	
Grading					
G1	77 (46.4%)	32 (3.8%)	28 (5.3%)	17 (6.6%)	
G2	1430 (86.1%)	721 (85.3%)	454 (85.8%)	255 (87.5%)	
G3	154 (92.7%)	92 (10.9%)	47 (8.9%)	15 (5.8%)	0.05
Missing	148	89	31	27	
Type of surgery					
Anterior/deep anterior resection	1169 (66.0%)	620 (66.4%)	367 (65.5%)	182 (65.9%)	
Intersphincteric resection	105 (5.9%)	63 (6.7%)	30 (5.4%)	12 (4.3%)	
Extirpation	445 (25.1%)	225 (24.1%)	142 (25.4%)	78 (28.3%)	
Other	51 (2.9%)	26 (2.8%)	21 (3.8%)	4 (1.4%)	0.276
No surgery	8	0	0	8	
Missing	0	0	0	0	
TME					
Complete	1405 (84.4%)	745 (86.7%)	430 (80.7%)	230 (84.6%)	
No complete	259 (15.6%)	114 (13.3%)	103 (19.3%)	42 (15.4%)	0.01
Missing	114	75	27	12	
ypT					
ypT0	311 (17.5%)	123 (13.2%)	113 (20.2%)	75 (26.4%)	
ypT1	120 (6.7%)	58 (6.2%)	38 (6.8%)	24 (8.5%)	
ypT2	519 (29.2%)	292 (31.3%)	157 (28.0%)	70 (24.6%)	
ypT3	776 (43.6%)	433 (46.4%)	238 (42.5%)	105 (37.0%)	
ypT4	52 (2.9%)	28 (3.0%)	14 (2.5%)	10 (3.5%)	<0.001
Missing	0	0	0	0	
ypN					
ypN0	1287 (72.4%)	666 (71.3%)	401 (71.6%)	220 (77.5%)	
ypN+	491 (27.6%)	268 (28.7%)	159 (28.4%)	64 (22.5%)	0.016
Missing	0	0	0	0	
ypL					
ypL negative	1545 (87.0%)	778 (83.4%)	505 (90.2%)	262 (92.9%)	
ypL positive	230 (13.0%)	155 (16.6%)	55 (9.8%)	20 (7.1%)	<0.001
Missing	1	1	0	0	
ypV					
ypV negative	1686 (94.4%)	877 (94.7%)	534 (95.4%)	275 (97.5%)	
ypV positive	82 (4.6%)	49 (5.3%)	26 (4.6%)	7 (2.5%)	0.145
Missing	10	8	0	2	
Resection status					
R0	1714 (98.1%)	913 (98.8%)	543 (97.8%)	258 (95.9%)	
R1 °	34 (1.9%)	11 (1.2%)	12 (2.2%)	11 (4.1%)	0.009
Missing	403	10	5	15	
TRG					
TRG 0/1	300 (17.9%)	188 (20.6%)	76 (13.9%)	36 (12.9%)	
TRG 2/3	1131 (64.0%)	604 (66.3%)	357 (65.4%)	170 (61.2%)	
TRG 4	304 (18.1%)	119 (13.1%)	113 (20.7%)	72 (25.9%)	<0.001
Missing	43	23	14	6	

Correlations were assessed using the Pearson’s chi-squared test * or Kruskal–Wallis test **; ° Patients with R2 resection have been excluded from this study. Abbreviations: CRT, chemoradiotherapy; TNT, total neoadjuvant treatment; TRG, Dworak tumor regression grading. Bold printed: significant *p* < 0.05.

**Table 2 cancers-16-03673-t002:** Competing risk regression model including downstaging, neoadjuvant treatment approach, and interaction term downstaging x treatment approach.

Competing Risk Regression Model	Treatment Failure		
	HR	CI 95%	*p*-Value
No response/progress	Reference		
Limited downstaging	0.37	(0.26–0.51)	<0.001
Intermediate downstaging	0.17	(0.11–0.28)	<0.001
Extensive downstaging	0.19	(0.09–0.39	<0.001
5-FU CRT	Reference		
5-FU/Ox CRT	0.81	(0.60–1.09)	0.16
TNT	1.01	(0.68–1.52)	0.95
Interaction terms			
Limited downstaging × 5-FU/Ox CRT	1.23	(0.69–2.19)	0.48
Limited downstaging × TNT	0.73	(0.28–1.91)	0.52
Intermediate downstaging × 5-FU/Ox CRT	0.80	(0.23–2.81)	0.73
Intermediate downstaging × TNT	1.29	(0.62–2.65)	0.50
Extensive downstaging × 5-FU/Ox CRT	1.10	(0.42–2.85)	0.85
Extensive downstaging × TNT	1.15	(0.38–3.46)	0.81

**Table 3 cancers-16-03673-t003:** Competing risk regression model including TRG, neoadjuvant treatment approach, and interaction term TRG × treatment approach.

Competing Risk Regression Model	Treatment Failure		
	HR	CI 95%	*p*-Value
TRG 4	Reference		
TRG 2/3	3.71	(1.84–7.49)	<0.001
TRG 0/1	6.48	(3.14–13.4)	<0.001
5-FU CRT	Reference		
5-FU/Ox CRT	1.03	(0.38–2.82)	0.95
TNT	1.60	(0.61–4.21)	0.34
Interaction terms			
TRG 2/3 × 5-FU/Ox CRT	0.85	(0.30–4.21)	0.76
TRG 2/3 × TNT	0.63	(0.20–1.95)	0.42
TRG 4 × 5-FU/Ox CRT	0.67	(0.24–1.86)	0.44
TRG 4 × TNT	0.41	(0.12–1.35)	0.14

## Data Availability

The data presented in this study are not available due to the data protection requirements of the trials.

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
