# Peer review of "Tumor Response and Its Impact on Treatment Failure in Rectal Cancer: Does Intensity of Neoadjuvant Treatment Matter?"

_cancers, 2024, doi:10.3390/cancers16213673_

Round 1
Reviewer 1 Report
Comments and Suggestions for Authors
The authors conducted a retrospective analysis on subgroups of patients enrolled in several randomized clinical trials, in order to evaluate: 1) the role of intensified neoadjuvant treatment on tumor response; 2) the role of tumor response after neoadjuvant treatment on treatment failure; 3) the role of intensified neoadjuvant therapy on treatment failure.
The topic is very interesting and the methods are very elegant, however the analysis has some incorrigible limitations:
· The definition of the outcome “tumor response” is unreliable, because it accounts for the difference between post treatment staging and pre-treatment staging of patients managed in different time periods and, probably, with different procedures (CT-scan, MRI, ultrasound…)
· Follow up procedures and timing can be different among the clinical trials, thus confounding the outcome “treatment failure”
Furthermore, even though the results are clearly shown, I cannot understand the conclusions reported at the end of the abstract and at the end of the paper.
The main results of the analysis are:
1) the intensified neoadjuvant treatment significantly increases tumor response, defined either as extensive downstaging (figure 1) or as pathological complete response (ypT0N0 – Figure 2);
2) the occurrence of pathological complete response (ypT0N0), whatever achieved (chemoradiation – Fig S5; oxa chemoradiation – fig S6; or TNT - fig S7) significantly reduces the incidence of treatment failure;
3) the occurrence of downstaging (defined as tumor response) significantly reduces the incidence of treatment failure (Fig S4);
4) the curves depicted in figure 4 (se Figure 2 line 158) show that the clinical outcome (defined as treatment failure) of patients who achieve ypT0N0/ypT+N0/ypanyTN+ is the same regardless of the therapy used to achieve the post-treatment stage.
In the discussion paragraph the authors hypothesize that a pathological complete response or a poor response obtained after TNT could have a different survival outcome as compared with a pathological complete response or a poor response obtained after capecitabine-based chemoradiation treatment. Why? A ypT0N0 has a good prognosis and a ypT3N+ a poor prognosis irrespective of the treatment used to achieve that stage. Whether TNT produces a longer survival and/or a decreased incidence of distant metastases as compared to chemoradiation can only be addressed within a randomized clinical trial. Furthermore, if/which adjuvant therapy is most appropriate after a poor/intermediate response to cape-chemoradiation or TNT, again can only be addressed within a randomized clinical trial.
I suggest to re-phrase the conclusion, focusing it on the results, and to reconsider the allocation of the figures between the paper and the supplementary material.
Some editing mistakes:
· Figure 1 and Figure 2 are repeated twice; perhaps the figure at line 125 has to be entitled as “Figure 3” and the figure at line 158 as “Figure 4”.

Author Response
Reviewer #1
The authors conducted a retrospective analysis on subgroups of patients enrolled in several randomized clinical trials, in order to evaluate: 1) the role of intensified neoadjuvant treatment on tumor response; 2) the role of tumor response after neoadjuvant treatment on treatment failure; 3) the role of intensified neoadjuvant therapy on treatment failure.
The topic is very interesting and the methods are very elegant, however the analysis has some incorrigible limitations:
1st comment: The definition of the outcome “tumor response” is unreliable, because it accounts for the difference between post treatment staging and pre-treatment staging of patients managed in different time periods and, probably, with different procedures (CT-scan, MRI, ultrasound…)
Reply: Thank you for your remark. We agree and have commented on this in the limitation section (Line 246-257).
2nd comment: Follow up procedures and timing can be different among the clinical trials, thus confounding the outcome “treatment failure”
Reply: Thank you for your comment. We agree and have discussed this in the limitations section (Line 246-257). Although one might suggest that this bias may have led to earlier detection of treatment failure in the most recent CAO/ARO/AIO-12 study, we were not able to prove our original hypothesis that ypN+ after TNT is associated with earlier or more treatment failure than ypN+ after 5-FU CRT alone even taking this bias into account.
3rt comment: Furthermore, even though the results are clearly shown, I cannot understand the conclusions reported at the end of the abstract and at the end of the paper.
The main results of the analysis are:
1) the intensified neoadjuvant treatment significantly increases tumor response, defined either as extensive downstaging (figure 1) or as pathological complete response (ypT0N0 – Figure 2);
2) the occurrence of pathological complete response (ypT0N0), whatever achieved (chemoradiation – Fig S5; oxa chemoradiation – fig S6; or TNT - fig S7) significantly reduces the incidence of treatment failure;
3) the occurrence of downstaging (defined as tumor response) significantly reduces the incidence of treatment failure (Fig S4);
4) the curves depicted in figure 4 (se Figure 2 line 158) show that the clinical outcome (defined as treatment failure) of patients who achieve ypT0N0/ypT+N0/ypanyTN+ is the same regardless of the therapy used to achieve the post-treatment stage.
In the discussion paragraph the authors hypothesize that a pathological complete response or a poor response obtained after TNT could have a different survival outcome as compared with a pathological complete response or a poor response obtained after capecitabine-based chemoradiation treatment. Why? A ypT0N0 has a good prognosis and a ypT3N+ a poor prognosis irrespective of the treatment used to achieve that stage. Whether TNT produces a longer survival and/or a decreased incidence of distant metastases as compared to chemoradiation can only be addressed within a randomized clinical trial. Furthermore, if/which adjuvant therapy is most appropriate after a poor/intermediate response to cape-chemoradiation or TNT, again can only be addressed within a randomized clinical trial.
I suggest to re-phrase the conclusion, focusing it on the results, and to reconsider the allocation of the figures between the paper and the supplementary material.
Reply: Thank you for these important comments. During the conduct of CAO/ARO/AIO-12 and our ACO/ARO/AIO-18.1 phase III trial, we conducted several discussions with our investigators regarding the management of patients with ypN+ following TNT. Concern was expressed that persistent lymph node metastasis after TNT might be associated with a "very" aggressive phenotype of rectal cancer, and unlike the 5-FU CRT era, no additional adjuvant chemotherapy was included in the trial protocols. The aim of this study was to analyse whether the "concerns" were justified, as we did not initially state that association of treatment response and prognosis is irrespectively of treatment intensity (doi:10.1002/cncr.29260). However, as we showed, ypN+ after TNT was not associated with a significant risk of treatment failure. We have reworded the conclusion and added the limitation that prospective evidence to accurately address these questions is lacking (Line 259-268). Considering our initial hypothesis and the response from the other reviewers we do not change the sequence of our figures.
4th comment: Some editing mistakes:
- Figure 1 and Figure 2 are repeated twice; perhaps the figure at line 125 has to be entitled as “Figure 3” and the figure at line 158 as “Figure 4”.
Reply: Thanks, we have corrected this.

Reviewer 2 Report
Comments and Suggestions for Authors
Interesting topic, most critical issue is missing data to support conclusions and provide the backgroud for interpretation.
Major issues:
Missing Data: pT, pN, pM, tumor size, tumor regression grading (Dworak)/histological (complete) response, lympho-/vascular invasion (L/V), perineural spread (Pn), histological grading (G), resection status (R), quality of TME (Quirke/MERCURY), type of resection (LAR/APR), type of relapse (local, liver, lung, other), BMI/ASA (and/or other clinical parameters, if known)
Missing data should be presented in clinical and pathological characteristics as main tables (not supplementary) and correlated with different neoadjuvant approaches and outcome measures (TF, DFS, OS). Conclusions must be adapted if neccesary.
Tumor response was defined as downstaging from pre-therapeutical clinical (cTNM) to post-therapeutical pathological (ypTNM) and clinical (ycTNM) stage. This approach is problematic due to the different data basis. The authors should critically discuss that issue and provide additional data (as outlined obove) with correlation to histological tumor response (TRG).
Why was TF defined differently in the prevoius paper (Diefenhardt JAMA Open 2023)? Particularly, why was R2 prevoiusly included in TF and not here? To facilitate comparison to the previous work, uniform definitions should be applied or differences discussed appropriately. Alternatively, the authors could perform a comparative analysis with the missing data.
Minor issues:
P2 L48-49 "Nevertheless, ypN+ after TNT was reported in 25% and 17% of patients treated in the RAPIDO and the PRODIGE23 trials, respectively [5,6]." Relation to TNT shoud be outlined more clearly to the reader.
P2 L81 "...respectively extent of treatment response" should probably read "...respectively with extent of..."
P3L110 "Kruskal-Wall Test" > "Kruskal-Wallis test"
Author Response
Interesting topic, most critical issue is missing data to support conclusions and provide the backgroud for interpretation.
Major issues:
1st comment: Missing Data: pT, pN, pM, tumor size, tumor regression grading (Dworak)/histological (complete) response, lympho-/vascular invasion (L/V), perineural spread (Pn), histological grading (G), resection status (R), quality of TME (Quirke/MERCURY), type of resection (LAR/APR), type of relapse (local, liver, lung, other), BMI/ASA (and/or other clinical parameters, if known)
Missing data should be presented in clinical and pathological characteristics as main tables (not supplementary) and correlated with different neoadjuvant approaches and outcome measures (TF, DFS, OS). Conclusions must be adapted if neccesary.
Reply: Thank for remark. We have rearranged Table S1 to include it in the main manuscript as Table 1 and added the parameters you requested. Please, note that information regarding the detailed location of distant metastasis (lung vs liver vs other) was not available for all patients in our study and, was therefore, not included in this analysis. Parameters were analysed regarding their association with TF. We decided not to use DFS as the definition differs between the trials and we did not use OS as we previous show a significant difference in survival after treatment failure between the cohorts. The latter highlights the limitation of OS as clinical endpoint for the assessment the efficacy of primary treatments in trial cohorts recruiting patients over a longer period of time.
Further we added a Supplementary Table (Supplementary Table S2) to present the association of clinical/surgical/pathological parameters with risk of TF founding nothing which changed our conclusions.
2nd comment: Tumor response was defined as downstaging from pre-therapeutical clinical (cTNM) to post-therapeutical pathological (ypTNM) and clinical (ycTNM) stage. This approach is problematic due to the different data basis. The authors should critically discuss that issue and provide additional data (as outlined obove) with correlation to histological tumor response (TRG).
Reply: Thank for remark. To address the effect of different neoadjuvant strategies in rectal cancer the differences in baseline clinical stage should be taken into consideration. However, these cannot be accurately reflected by the pathological TRG. We have described this point in the limitations of our analysis (Line246-257)
3rd comment: Why was TF defined differently in the prevoius paper (Diefenhardt JAMA Open 2023)? Particularly, why was R2 prevoiusly included in TF and not here? To facilitate comparison to the previous work, uniform definitions should be applied or differences discussed appropriately. Alternatively, the authors could perform a comparative analysis with the missing data.
Reply: Thank you for this important remark. We decided to include only patients with curative post-surgical status in the present analysis. Patients with metastases during treatment and those with R2 status were excluded as these patients are receiving individual salvage treatments. Figure S2 provides the exact number of patients being excluded from this study according to the neoadjuvant treatment they received. To note, the relative number of patients is relatively small and comparable for all treatment strategies (5-FU CRT 8.1%, 5-FU/OX CRT 7.7%, TNT 7.2%).
4th comment: Minor issues: P2 L48-49 "Nevertheless, ypN+ after TNT was reported in 25% and 17% of patients treated in the RAPIDO and the PRODIGE23 trials, respectively [5,6]." Relation to TNT shoud be outlined more clearly to the reader.
Reply: Thank you, we have rephrased this sentence and included the percentage of ypN+ in the standard of care group.
5th comment: P2 L81 "...respectively extent of treatment response" should probably read "...respectively with extent of..."
Reply: Thank you, we have rephrased this sentence for clarification.
6th comment: P3L110 "Kruskal-Wall Test" > "Kruskal-Wallis test"
Reply: Thank you, this has been corrected accordingly.

Reviewer 3 Report
Comments and Suggestions for Authors
This paper is a comprehensive analysis of outcomes of various treatment strategies for patients with operable rectal cancer and compares the results from 3 studies - the topic is highly clinical relevant as we see a rising incidence of these cancers in young patients.
I struggled to understand the conclusions of the study in both the lay and non lay summary - the authors appear to be saying that despite higher response rates newer more intense regimens don't improve patient outcomes ? The lay summary should be understandable for a patient advocate at a minimum.
It would be helpful to contextualise these conclusions in the paper - what do these observations suggest for further studies and research directions ? should oxaliplatin based regimens be abandoned in this neoadjuvant setting if higher response rates ( and greater quality of survival impacting toxicity ) don't meaningfully affect outcomes ? what are the groups direction of research in light of these findings ?
the methodology , results , references are all clearly stated and appropriate - the narrative around the findings needs to be clearer as outlined above
Comments on the Quality of English Language
as-sociation should be association
other comments as outlined above
Author Response
This paper is a comprehensive analysis of outcomes of various treatment strategies for patients with operable rectal cancer and compares the results from 3 studies - the topic is highly clinical relevant as we see a rising incidence of these cancers in young patients.
1st comment: I struggled to understand the conclusions of the study in both the lay and non lay summary - the authors appear to be saying that despite higher response rates newer more intense regimens don't improve patient outcomes ? The lay summary should be understandable for a patient advocate at a minimum.
Reply: Thanks for your important remark. We have rephrased the summary per your request.
2nd comment: It would be helpful to contextualise these conclusions in the paper - what do these observations suggest for further studies and research directions ? should oxaliplatin based regimens be abandoned in this neoadjuvant setting if higher response rates ( and greater quality of survival impacting toxicity ) don't meaningfully affect outcomes ? what are the groups direction of research in light of these findings ?
Reply: Thank you for your question. The rationale for the present analysis was generated during the recruitment phase of our ACO/ARO/AIO-18.1 ongoing phase III trial (TNT per RAPIDO: vs. TNT per CAO/ARO/AIO-12/OPRA trials; primary endpoint: organ preservation rate). Indeed, it remains unclear which is the best way to treat patients that have responded poorly to TNT. Our hypothesis was that these patients without an adequate response to TNT were at very high risk of treatment failure. We hypothesized that tumors that did not respond to TNT reflected aggressive rectal cancer phenotypes. As expected, ypTNM stage was associated with the risk of treatment failure, but unexpectedly poor response to TNT was not associated with an extensive risk of TF than after CRT. Therefore, poor responders to CRT might benefit from curative surgery but not additional chemotherapy, whereas intensified surveillance schedule might be clinically relevant in this patient subgroup.
3rd comment: the methodology , results , references are all clearly stated and appropriate - the narrative around the findings needs to be clearer as outlined above
Reply: Thank you. We have rephrased the simple summary and the conclusion section considering the feedback from the other reviewers.

Round 2
Reviewer 2 Report
Comments and Suggestions for Authors
To address the effect of different neoadjuvant strategies in rectal cancer the differences in baseline clinical stage should be taken into consideration. However, these cannot be accurately reflected by the pathological TRG. We have described this point in the limitations of our analysis (Line246-257)”
It is reasonable to include baseline characteristics to assess treatment effects, but the comparison of clinical and pathological stage represents an inherent methological flaw that has not been clearly addressed in L246-257 or elsewhere in the paper (and must be discussed). It is particularly challenging to use imaging as a baseline for comparison with ypN: https://pubmed.ncbi.nlm.nih.gov/25638800/
Of course, TRG has its own limitations (that must be discussed!) but it represents a well-recognized independent parameter of tumour response and as such would be of great interest to the reader for evaluating the hypotheses of the paper. Please provide competing risk regression analyses stratified by TRG as in Figures S4 through S7 and as in Table 2.
Minor issues:
Materials and Methods: Statistical tests are not accurately described and referenced, especially for the competing risk regression analysis. Definition for TRG is missing. Please provide appropriate amendments and references.
Table 1: Correlations are not individually specified with the groups (as in table S2) and should be amended for consistency.
Figure 4: Specifications of the plots should be labeled within the figures for clarity and the plots should be distinguished as 4a/b/c.
Rebuttal: “We decided not to use DFS as the definition differs between the trials and we did not use OS as we previous show a significant difference in survival after treatment failure between the cohorts. The latter highlights the limitation of OS as clinical endpoint for the assessment the efficacy of primary treatments in trial cohorts recruiting patients over a longer period of time.”
Please include these remarks and appropriate references in the paper.
Editing:
L253: “a post-treatment”
L255 “recommendations”
Table S2: Headline “cli.nical”, tumor location “Lowe”
In all headlines of the Figures S3 – S7 the word “Incidence” was accidentally capitalized.
Author Response
Reviewer #2
1st comment: To address the effect of different neoadjuvant strategies in rectal cancer the differences in baseline clinical stage should be taken into consideration. However, these cannot be accurately reflected by the pathological TRG. We have described this point in the limitations of our analysis (Line246-257)”
It is reasonable to include baseline characteristics to assess treatment effects, but the comparison of clinical and pathological stage represents an inherent methological flaw that has not been clearly addressed in L246-257 or elsewhere in the paper (and must be discussed). It is particularly challenging to use imaging as a baseline for comparison with ypN: https://pubmed.ncbi.nlm.nih.gov/25638800/
Reply: Thanks for this important comment. We have added two sentences to our limitations section to reflect your thoughts (Line 318-322). Unfortunately, imprecision in clinical staging is a problem in daily clinical practice or clinical research in rectal cancer, although you might consider that this problem should be balanced in randomised trials (RAPIDO, Prodige23, PROSPECT). Thank you very much for this important feedback.
2nd comment: Of course, TRG has its own limitations (that must be discussed!) but it represents a well-recognized independent parameter of tumour response and as such would be of great interest to the reader for evaluating the hypotheses of the paper. Please provide competing risk regression analyses stratified by TRG as in Figures S4 through S7 and as in Table 2.
Reply: Thank you very much. We have added a short sentence about the limitations of TRG in this study (Line 322-324). The results of the competing risk regression with TRG stratified by treatment approach are reported in the last lines of Table S2 and as Table S3 (Table S3 is the TRG equivalent of Figure 4). The additional results of the competing risk regression model with TRG are reported in Table 3, and the results of the moderation effect analysis by TRG have been added to the manuscript (no significance, Line245-250).
3rd comment: Materials and Methods: Statistical tests are not accurately described and referenced, especially for the competing risk regression analysis. Definition for TRG is missing. Please provide appropriate amendments and references.
Reply: We have added the definition of TRG (Line 79 – 84) and added two short paragraphs to explain competing risk regression (Line 105-109) and moderation effect analysis (97-101) with references.
4th comment: Table 1: Correlations are not individually specified with the groups (as in table S2) and should be amended for consistency.
Reply: We have added the numbers and percentages of the distribution in the full study cohort in Table 1. However, the correlation should remain as intended to describe differences in the distribution of clinical, surgical, and pathological characteristics between different neoadjuvant treatments and not within a treatment group.
5th comment: Figure 4: Specifications of the plots should be labeled within the figures for clarity and the plots should be distinguished as 4a/b/c.
Reply: Figure 4 has been changed to reflect your comment.
6th comment: Rebuttal: “We decided not to use DFS as the definition differs between the trials and we did not use OS as we previous show a significant difference in survival after treatment failure between the cohorts. The latter highlights the limitation of OS as clinical endpoint for the assessment the efficacy of primary treatments in trial cohorts recruiting patients over a longer period of time.” Please include these remarks and appropriate references in the paper.
Reply: We have added this paragraph in the Methods section and added the reference to our previous work (Line 87-91).
Additional comments: Editing: L253: “a post-treatment” – corrected, L255 “recommendations” – corrected, Table S2: Headline “cli.nical” – not found, tumor location “Lowe” – corrected, In all headlines of the Figures S3 – S7 the word “Incidence” was accidentally capitalized – corrected.
Reviewer 3 Report
Comments and Suggestions for Authors
My comments raised in the initial review have been addressed - thank you
Comments on the Quality of English Languageas-sociation should be association
Author Response
Comments and Suggestions for Authors
My comments raised in the initial review have been addressed - thank you
We thank you for taking the time to review our work and appreciate your input.
Round 3
Reviewer 2 Report
Comments and Suggestions for Authors
Thanks to the authors for their efforts with the revision. TRG-equivalents of Figures S4 to S7 still missing.
editing:
L99: "what can help" > ,which can help
L222 ff: Figure 44a, b, c > Figure 4a, b, c
first, second third plot (in caption) > a, b, c
L311: "pathological is bias" > pathological stage is biased
Author Response
Comment 1. Thanks to the authors for their efforts with the revision. TRG-equivalents of Figures S4 to S7 still missing.
Response: We have added the requested figures as Figures S8-S11. We have also added the TRG equivalents of Figure 4 as Figures S12-S14. The hazard ratios have already be reported in Table S2 and Table S3.
Comment 2 editing: L99: "what can help" > ,which can help L222 ff: Figure 44a, b, c > Figure 4a, b, c first, second third plot (in caption) > a, b, c L311: "pathological is bias" > pathological stage is biased
Response: Thanks, we have updated the manuscript.